# Identification of a *Cordyceps fumosorosea* Fungus Isolate and Its Pathogenicity against Asian Citrus Psyllid, *Diaphorina citri* (Hemiptera: Liviidae)

**DOI:** 10.3390/insects13040374

**Published:** 2022-04-11

**Authors:** Yaru Luo, Shujie Wu, Xinyu He, Desen Wang, Yurong He, Xiaoge Nian

**Affiliations:** Department of Entomology, College of Plant Protection, South China Agricultural University, Guangzhou 510642, China; luoyaru1114@163.com (Y.L.); wushujie2021@163.com (S.W.); w2080505078@163.com (X.H.); desen@scau.edu.cn (D.W.)

**Keywords:** HLB, citrus greening, entomopathogen, biological control

## Abstract

**Simple Summary:**

Some entomopathogenic fungi are highly pathogenic to *Diaphorina citri*, which is the mainly transmitting vector of *C*Las. In our study, we isolated an entomopathogenic fungus strain from an adult cadaver of *D. citri*. The isolate was identified as *Cordyceps fumosorosea*, based on morphology and ITS sequence analysis, and named *C. fumosorosea* SCAU-CFDC01. We further evaluated its pathogenicity against *D. citri* nymphs and adults under laboratory and greenhouse conditions. The laboratory results showed that *C. fumosorosea* SCAU-CFDC01 was most pathogenic to young nymphs, followed by old nymphs and adults. Furthermore, the results on greenhouse experiments revealed that *C. fumosorosea* SCAU-CFDC01 not only had high pathogenicity against nymphs, but also had serious negative effects on adults from nymphs treated, including emergence rate of adults and female longevity. Our results showed *C. fumosorosea* SCAU-CFDC01 was highly pathogenic to *D. citri*, and these findings will facilitate mycoparasite development for biological control of *D. citri*.

**Abstract:**

*Diaphorina citri* is the mainly transmitting vector of the citrus huanglongbing pathogen, which causes severe losses in in the citrus industry. In this study, we isolated a new entomopathogenic fungus, identified as member of *Cordyceps fumosorosea* based on morphology and ITS sequence analysis. We named *C. fumosorosea* SCAU-CFDC01 and evaluated its pathogenicity against *D. citri* nymphs and adults by immersion under laboratory and greenhouse conditions. Results showed that SCAU-CFDC01 was most pathogenic to young nymphs, followed by old nymphs and adults. The LC_50_ values of the fungus on nymphs and adults showed a declining trend over a 2–7-day period after inoculation. The LT_50_ (lethal time for a certain concentration to cause 50% mortality) values also presented a decreasing trend along with increasing conidia concentrations. For the results on greenhouse experiments, when 3rd and 5th instar nymphs were inoculated with 1 × 10^5^ conidia mL^−1^, the survival rate of nymphs were lower, and the emergence rate of adults and female longevity was significantly reduced compared with the control. However, there were no significant effects on sex ratio of adults and male longevity. Our results showed SCAU-CFDC01 was highly pathogenic to *D. citri*, and may promote mycoparasite development for biological control of *D. citri*.

## 1. Introduction

Huanglongbing (HLB), also known as citrus greening disease, is a highly severe citrus disease [1,2,3]. After citrus is infected with HLB, the typical symptoms include yellow mottling on leaves and shoots, stunted growth, premature fruit drop, small and green fruit, and so on [1,4]. There are three species of Gram-negative, phloem-limited bacteria associated with HLB symptoms: *Candidatus* Liberibacter asiaticus (*C*Las), *Candidatus* L. africanus (*C*Laf), and *Candidatus* L. americanus (*C*Lam). *C*Las and *C*Lam are transmitted by *Diaphorina citri*, while *C*Laf is transmitted by *Trioza erytreae* [1]. In China, HLB is transmitted by *D. citri* in a propagative–circulative manner [5,6]. As a result, control of *D. citri* is the most effective measure to prevent HLB epidemics.

To date, chemical control is still the principal management strategy for *D. citri*. However, injudicious usage of chemical pesticides had led to a series of adverse environmental issues, such as pesticide resistance of insects, harm to beneficial insects, and insecticide residues [7,8]. In addition, insecticides are restricted for urban areas where *D. citri* may reproduce on ornamental (i.e., *Murraya paniculata* (L) Jacks plants) and backyard citrus plants. Moreover, *D. citri* has been reported to develop resistance to at least two pesticides (malathion and chlorpyrifos) [9]. Therefore, alternative management methods such as biological control have been gradually introduced. Entomopathogenic fungi, as an important agent of biological control, play a crucial role. Once touch with the insect epidermis, the fungus multiplies within the insect eventually killing the host, then emerges and sporulates on the cadaver. Finally, the fungus breaks through the epidermis and releases conidiophores to infect other insects [10,11,12].

Some entomopathogenic fungi are highly pathogenic to phloem-sucking pests such as aphids, thrips, and plant hoppers [13]. Many species, including *Hirsutella citriformis*, *Metarhizium anisopliae*, *Cordyceps bassiana*, *Beauveria bassiana*, *Cordyceps javanica*, *C. fumosorosea* (*Isaria fumosorosea*), and *Clonostachys rosea*, are reported to be highly pathogenic to *D. citri* [14,15,16,17,18,19]. For example, after application 4 times using 2 × 10^13^ conidia ha^−1^ of *C. fumosorosea*, *B. bassiana*, and *M. anisopliae* under field conditions, the mortality rates of *D. citri* were 40, 60, and 50% in nymphs, and 42, 50, and 50% in adults, respectively [15]. *M. anisopliae* and *B. bassiana* strains caused a more serious infection in the nymphs and adults of *D. citri* in comparison with *C. fumosorosea* isolates [20]. Nevertheless, few isolates targeting *D. citri* have been found in China. Ou et al. [17] reported that *C. javanica* GZQ-1 isolate was highly pathogenic to *D. citri*, and resulted in 91.7%, 88.3%, 73.3%, and 72.2% mortality rates of younger instar, median instar, older instar nymphs, and adults, respectively. Awan et al. [18] tested 12 fungal isolates and found that *C. javanica* FHY002-ACPHali was the most pathogenic to *D. citri*.

In the current study, a new isolate of *C. fumosorosea* was cultured from a cadaver of *D. citri* in the citrus-producing areas in Zhanjiang city, Guangdong Province, China. The aims of this study were the following: (ⅰ) to identify the isolate using molecular and morphological characteristics; (ⅱ) to investigate its pathogenicity to nymphs and adults of *D. citri* under laboratory conditions; (ⅲ) to evaluate the effects of the isolate on emergence rate, sex ratio, and longevity of adults from nymphs treated in the greenhouse. The overall objective was to determine whether the isolate could be used as a potential mycoparasite for the control of *D. citri*.

## 2. Materials and Methods

### 2.1. Sampling and Isolation of Entomopathogenic Fungus

Firstly, the entomopathogenic fungus was isolated from an adult cadaver of *D. citri* from the citrus-producing area in Zhanjiang City of Guangdong Province, China. The cadaver was surface-sterilized by immersion in 75% ethanol, and rinsed at least three times using sterile distilled water. After drying on filter paper, the cadaver was incubated on water medium in an incubator (4 ± 1 °C, 65 ± 5% RH (relative humidity) with 24 h total darkness). Once small mycelia from the cadaver were observed, the fresh fungal colony was picked and subcultured on SDAY medium a couple of times to obtain the fungal isolate. The fungal isolate was stored in the Key Laboratory of Bio-Pesticide Innovation and Application, South China Agricultural University (SCAU), China.

### 2.2. Morphological Characterization and Inoculation

The isolate was inoculated on SDAY medium and incubated at 26 ± 1 °C, 65 ± 5% RH with a 14 light (L):10 dark (D) h photoperiod. After 12 days inoculation, the conidia were scraped off from the plates, suspended into 0.05% Tween-80 (*v*/*v*), quantified, and pictures were taken under a Nikon eclipse 80i phase contrast microscope (Nikon, Tokyo, Japan). In addition, after 5th instar nymphs of *D. citri* were inoculated with 1 × 10^7^ conidia mL^−1^, the preliminary morphology was recorded on the 3rd, 5th, and 9th day, and identified using the taxonomy keys of Humber [21,22]. In addition, the mycelia from the cadaver were picked and subcultured on SDAY medium for completing the Koch postulates.

### 2.3. Indentification and Phylogenetic Tree of Fungal Isolate

For extraction of genomic DNA (gDNA), the isolate was grown on SDAY medium at 26 ± 1 °C for 7 days. The mycelia were harvested, and the gDNA was extracted using a fungal DNA kit following the manual (Sangon Biotech, Shanghai, China). To amplify and sequence the internal transcribed spacer (ITS) of the isolate, the universal primers ITS1 (5′-TCCGTAGGTGAACCTGCGG-3′) and ITS5 (5′-GGAAGTAAAAGTCGTAACAAGG-3′) were used. Total reaction volume of PCR amplification was 50 μL including 25 μL of PrimerSTAR^®^Max DNA Polymerase (Takara Biomedical Technology (Beijing) Co., Ltd., Beijing, China), 2 μL 10 mM ITS1, 2 μL 10 mM ITS5, 2 μL of total DNA template (76.9 ng/μL), and 19 μL of ddH_2_O. The thermal cycle conditions used in the PCR were: 95 °C for 5 min, followed by 35 cycles of 95 °C for 30 s, 60 °C for 30 s, 72 °C for 1 min, and 1 cycle of 72 °C for 5 min. PCR products were determined by 1% agarose gel electrophoresis and sent to the Sangon Biotech (Shanghai) for complete bidirectional sequencing with PCR primers. To determine the homogeneous counterparts, the DNA sequence of the isolate was submitted to the Basic Local Alignment Search Tool (BLAST) and aligned with other GenBank sequences (Table 1). For the phylogenetic analysis, the ITS sequences of the isolate and other related fungal species were selected by the Convention of Biological Diversity (CBD). Subsequently, a phylogenetic tree was built based on the maximum likelihood method using MEGA7 software. The clade stability of supporting values of the branches was based on bootstrap analysis with 1000 replicates.

### 2.4. Host Plants and Insect Colony

A total of 300 healthy *M. paniculata* plants were cultured in 40 cm-diameter plastic pots in the greenhouse in autumn of 2021. The *D. citri* colony was collected from the SCAU campus and reared on *M. paniculata* plants, which were individually covered with insect-rearing sleeves (50 × 45 cm). The plant colony was raised in an incubator (26 ± 1 °C, 65 ± 5% RH with a 14 L:10 D h photoperiod).

### 2.5. Laboratory Bioassays

A total of 3 assays were conducted to evaluate the pathogenicity of the isolate to *D. citri* nymphs (3rd and 5th instar nymphs) and the newly emerged adults (1–2 days old) under laboratory conditions. To obtain different instar nymphs of insects, 9 *M. paniculata* plants with flushes were moved into 4 fine-mesh screen cages (60 × 60 × 60 cm). Approximate 2000 *D. citri* adults of mixed gender were introduced in each cage for oviposition for 24 h. After 24 h, the adults were removed, and the plants with eggs were maintained in a growth incubator (26 ± 1 °C, 65 ± 5% RH with a 14 L: 10 D h photoperiod) for egg hatching and nymphs’ development. Conidia were suspended in 0.05% Tween-80 (*v*/*v*) and diluted with distilled water. Every assay consisted of 5 corresponding conidia concentrations (1 × 10^3^, 1 × 10^4^, 1 × 10^5^, 1 × 10^6^, 1 × 10^7^ conidia mL^−1^ for 3rd or 5th instar nymphs; 1 × 10^6^, 5 × 10^6^, 1 × 10^7^, 5 × 10^7^, 1 × 10^8^ conidia mL^−1^ for adults) plus a control check (CK) with 0.05% Tween-80. For inoculation, excised flushes with specific nymphs were dipped into above conidia concentrations and negative control for 30 s, and then air-dried at room temperature. After completion, each flush was individually put into 2.0 mL microcentrifuge tubes (Fisher Scientifific, Pittsburg, PA, USA) containing a 2% sucrose solution, and were placed into 50 mL microcentrifuge tubes sealed with gauze. Adults were fumigated with CO_2_, loaded on the gauze, immersed in above the concentrations for 30 s, and then transferred to flushes, which were put into clear plastic cups (30 × 15 cm) sealed with gauze. For all assays, there were 3 independent replicates (20–30 nymphs and adults per replicate) in each concentration treatment. The entire bioassay studies were repeated three times. All the devices and treated insects were maintained in the incubator (26 ± 1 °C, 65 ± 5% RH with a 14 L: 10 D h photoperiod). All treated nymphs and adults were checked daily for 7 days, and fresh untreated flushes were added as requirement. Dead insects from all treatments were transferred to moist filter paper for further fungal incubation.

### 2.6. Greenhouse Bioassays

Based on the above experiments, effects of the isolate against *D. citri* adults emerging from treated nymphs were investigated in the greenhouse. *M. paniculata* plants with about 10 fresh flushes were grown in 15 cm-diameter pots in the greenhouse. Approximately 300 *D. citri* adults of mixed gender were released per plant, and laid eggs for 12 h; they were then removed. Nine plants were introduced into one chamber (120 × 120 × 120 cm) with *D. citri*-proof meshes. When 3rd or 5th instar nymphs were observed, the number of nymphs per plant was counted (about 30–60 nymphs plant^−1^ observed). The 1 × 10^5^ conidia mL^−1^ suspension was sprayed on both sides of *M. paniculata* plants, at 20 mL plant^−1^ using a manual sprayer (SW-168 model). A measure of 0.05% Tween-80 was included as blank control. 

Nymphs per individual flush per plant were counted every day until emergence, and were grown from emergence under the stereomicroscope. Survival rate of nymphs, emergence rate, longevity, and sex ratio of adults from nymphs treated were recorded. Each treatment included three biological replicates, and the whole experiment was repeated three times on different days in autumn 2021.

### 2.7. Statistical Analyses

All the data were normalized prior to any data analysis. Corrected mortality was used based on Abbott’s formula [23]. For determining significant differences of insect mortality rate in the laboratory bioassay, split-plot ANOVA and Tukey’s honest significant difference (HSD) tests were carried out. The LT_50_, LC_50_, and their 95% confidence limits (CIs) were derived from probit analysis. Two-tailed paired Student’s *t*-test was used for analyzing the difference of development parameters (survival rate of nymphs, emergence rate, longevity, and sex ratio of adults from nymphs treated) between the control and fungal treatment in the greenhouse bioassays. SPSS 22.0 [24] and GraphPad Prism 9.0 software were used to perform all the statistical analyses.

## 3. Results

### 3.1. Identification and Characterization of the Isolate

The hyphae were white and velvet-like (Figure 1a). After 15 days of incubation, the surface of the culture medium was covered with mature smoke rose conidia (Figure 1b). The mature conidia were club-shaped (Figure 1c). After 5th instar nymphs of *D. citri* were infected with 1 × 10^7^ conidia mL^−1^, the nymphs were observed to be less active, moving more slowly, and began to die 24 h after inocubation. A duration of 48 h after inoculation, white mycelia were found in the leg, mouthparts, and intersegmental regions of the nymphs. After 3 days of inoculation, the white mycelia were found, and the whole body was covered with white mycelia on the 5th day. Smoke rose conidia were produced on the 9th day (Figure 1d–f).

The result of the morphological identification demonstrated that this isolate is *C. fumosorosea* [21,22]. Genomic analysis further confirmed the isolate has 100% homology with other *C. fumosorosea* strains, such as *C. fumosorosea* FAFU-1 (GenBank accession number MG837716.1), *C. fumosorosea* ARSEF: 3302 (HM209050.1), *C. fumosorosea* CNRCB1(HM209049.1), *C. fumosorosea* SKCH-1 (HM209050.1), and *C. fumosorosea* ifTS02 (KX057373.1) (Figure 2). We named the *C. fumosorosea* isolate SCAU-CFDC01 with the GenBank accession number OL872288.1.

### 3.2. Pathogenicity of C. fumosorosea SCAU-CFDC01 against D. citri

The total mortality trends of *D. citri* nymphs and adults in response to various concentrations of *C. fumosorosea* conidia over time after inoculation are illustrated in Figure 3. The mortality rates of both nymphs and adults were significantly affected by the interaction of conidia concentration and days after inoculation (3rd instar nymphs, and 5th instar, *p* < 0.001; adults, *p* < 0.05), meaning that inoculum load effect was dependent on the exposure time.

The LC_50_ values for nymphs and adults treated with various conidia concentration are shown in Table 2. Based on LC_50_ values, 3rd instar nymphs were the most susceptible to SCAU-CFDC01, followed by 5th instar nymphs, and then adults.

The LT_50_ values estimated decreased with increasing fungal concentrations, and the LT_50_ differences among three assays tended to diminish with increasing fungal concentrations. The LT_50_ estimates for 3rd instar nymphs, 5th instar nymphs, and adults were 6.93–1.18 days, 7.60–1.51 days, and 9.71–1.20 days, respectively, in the range 1 × 10^3^–1 × 10^8^ conidia mL^−1^ (Table 3). These results demonstrated that our *C. fumosorosea* isolate was more pathogenic against nymphs of *D. citri* than adults. The bioassay results indicated that *C. fumosorosea* SCAU-CFDC01 isolate was highly pathogenic to *D. citri*.

### 3.3. Greenhouse Experiments

After treatment with 1 × 10^5^ conida mL^−1^, the survival rates of adults from 3rd and 5th instar nymphs were only 9.20 ± 1.51% and 33.53 ± 3.19% and were significantly lower than the control (*p* < 0.001) (Figure 4a,b). Emergence rates of adults from 3rd and 5th nymphs treated were only 7.17 ± 1.21% and 26.32 ± 2.62%, respectively, which was significantly lower than the control groups (87.75 ± 2.29% and 91.24 ± 2.06%, respectively; *p* < 0.001) (Figure 4c,d). In addition, *C. fumosorosea* SCAU-CFDC01 significantly reduced female longevity, from 19.48 ± 0.92 days to 16.45 ± 0.84 days (*p* < 0.05), and from 20.61 ± 0.77 days to 16.47 ± 0.38 days (*p* < 0.001) for females from 3rd and 5th instar nymphs treated, respectively (Figure 4e,f). However, compared to the control, the isolate had no significant effects on male longevity and sex ratio of adults from 3rd/5th instar nymphs treated (*p* > 0.05) (Figure 4e–h).

## 4. Discussion

In this study, morpho-molecular characterization was performed to identify a new fungus isolate from *D. citri*. Firstly, the morphological proofs are consistent with the previously reported *C. fumosorosea* by Gallou et al. [25]. Next, BLAST analysis showed that it was 100% sequence homology with Cordyceps spp., and this result was further proved by phylogenetic analysis based on ITS1/ITS5 sequences (Figure 2).

To date, only a few entomopathogenic fungi have been shown to affect *D. citri*. Hoy et al. [26] isolated a new *C. fumosorosea* IfrAsCP and assessed its pathogenicity against *D. citri* with respective LT_50_ values of 4.63 and 4.27 days after the exposure to 1 × 10^7^ and 1 × 10^8^ conidia mL^−1^, respectively. Pinto et al. [27] reported that *B. bassiana* IBCB 66 was highly pathogenic to *D. citri* nymphs and the LC_50_ value on the 10th day after inoculation was 4.0 × 10^5^ conidia mL^−1^. Ibarra-Cortes et al. [20] showed that both isolates of *B. bassiana* and *M. anisopliae* caused the greatest infection in adults and nymphs of *D. citri*. Ou et al. [17] investigated the virulence of *C. javanica* GZQ-1 fungus to *D. citri*, in which LC_50_ and LT_50_ values against lower instar, medium instar, older instar, and adults were 1.20 × 10^5^, 1.10 × 10^6^, 4.47 × 10^6^, and 8.12 × 10^6^ conidia mL^−1^ and 4.25, 4.51, 5.17, and 5.49 days, respectively. In our current study, after inoculation with 1 × 10^7^ conidia mL^−1^, the mortality of younger and older instar nymphs achieved 100% mortality on the 3rd day, indicating that *C. fumosorosea* SCAU-CFDC01 isolate was better than other fungus described above terms of the pathogenicity to *D. citri* nymphs under controlled conditions.

In our study, 7 days after inoculation 1 × 10^7^ conidia mL^−1^, the mortality rate of adults was 66%. However, Stauderman et al. [14] reported that the mortality rates of *D. citri* adults reached 100% between 10^6^ and 10^7^ conidia mL^−1^ of *C. fumosorosea* for 12 days. Compared with the dipping method we used, *D. citri* adults continually in contact with conidia-contaminated grapefruit leaves for 12 days in Stauderman et al. [14] showed higher mortality rates. Similar results were reported by Orduño-Cruz et al. [16] and Ibarra-Cotres et al. [20]. In addition to the applied method, leading to difference in fungal virulence, numerous studies have revealed the pathogenicity of entomopathogenic fungi is also easily affect by various biotic and abiotic factors, such as host, light, humidity, temperature, and agricultural practices, etc., see [18,28,29,30,31]. Therefore, exploring the pathogenicity of potential fungi on different developmental stages and environmental conditions is important. Validation at field conditions is also fundamental.

Nymphs are more susceptible to fungal infection than adults of *D. citri*. Furthermore, nymphs are more like gregarious on young leaflets and move more slowly than adults, which means they are more fitting targets for biological control [32]. In addition, lethality requires an infection period; therefore, an infected adult may be able to transmit *C*Las before its eventual death. In the present study, along with the increasing in *D. citri* developmental stage, the corresponding LT_50_s and LC_50_s also increased when inoculated with the same conidia concentration, indicating that the *C. fumosorosea* SCAU-CFDC01 strain is more pathogenic to nymphs than adults, which is in accordance with previously published works [17,20]. However, we cannot ignore the adults, because they are more active in disseminating *C*Las. Bamisile et al. [33] assessed the pathogenicity of two fungal strains of *B. bassiana* and one strain of *I. fumosorosea* against adults of *D. citri* in a greenhouse setting, and found that there was a 50% reduction in the survival rate of *D. citri* adults within 5 days of exposure. Yang et al. [34] reported that the mortality rates of *D. citri* adults reached 46.7% mortality at 1 × 10^8^ conidia mL^−1^ of *Clonostachys rosea* for 9 days in the laboratory conditions. In our study, we also assessed the pathogenicity of SCAU-CFDC01 to adults and found 88% mortality at 1 × 10^8^ conidia mL^−1^ in the laboratory conditions. Furthermore, the results of greenhouse experiments showed that SCAU-CFDC01 not only had high virulence to nymphs, but also had serious negative effects on adults which emerged from treated nymphs. The survival rate of nymphs, emergence rate of adults and female longevity significantly decrease, which further revealed the potential good control effect of our fungus on *D. citri* population. In conclusion, our results and previously reported results findings [3,14,15,17,19,32,35] all showed that entomopathogenic fungi are excellent prospective options for the biological control of *D. citri*.

## 5. Conclusions

The discovery of entomopathogenic fungi will promote the development of biopesticides for the biological control of *D. citri*. In the current study, we identified a highly pathogenic isolate, *C. fumosorosea* SCAU-CFDC01, based on morphology and molecular proofs, and estimated its effect on the mortality, survival, emergence rate, longevity, and sex ratio of *D. citri*. Our study not only enriches the available resource library of entomopathogenic fungi, but also provides an alternative option for *D. citri* control.

## Figures and Tables

**Figure 1 insects-13-00374-f001:**
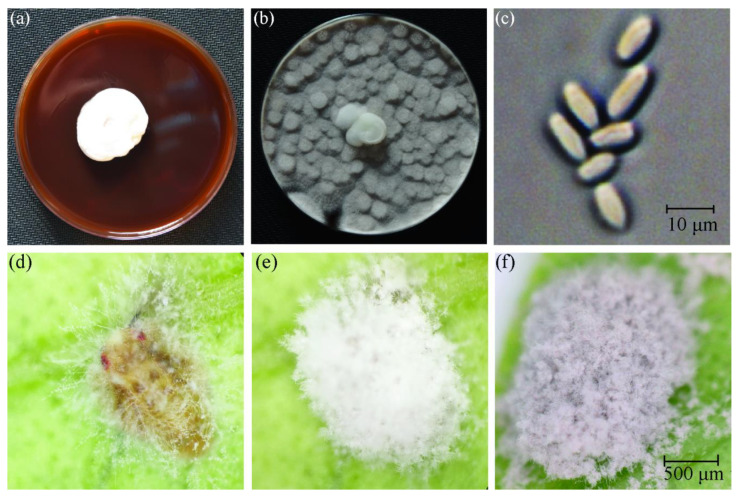
Morphology characteristics of the isolate on SDAY medium and 5th instar nymph of D. citri. (**a**) The upper side of the colony on SDAY on 5th day at 25 °C; (**b**) the upper side of the mature colony on SDAY on the 15th day at 25 °C; (**c**) mature conidia; (**d**) 5th instar nymph of *D. citri* inoculated with 1 × 10^7^ conidia mL^−1^ on 3rd day after inoculation; (**e**) 5th instar nymph of *D. citri* inoculated with 1 × 10^7^ conidia mL^−1^ on 5th day after inoculation; (**f**) 5th instar nymph of *D. citri* inoculated with 1 × 10^7^ conidia mL^−1^ on 9th day after inoculation.

**Figure 2 insects-13-00374-f002:**
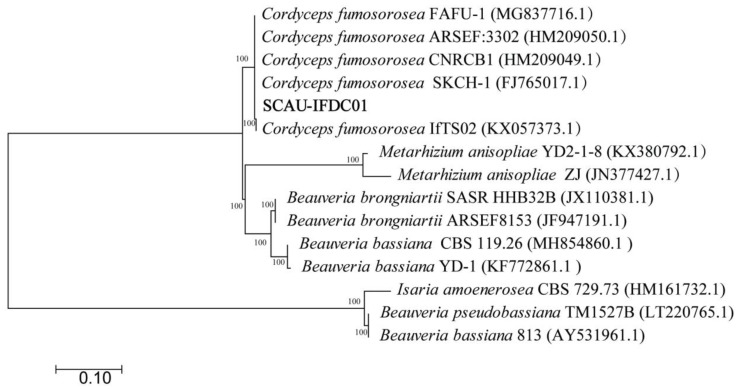
Molecular identification and phylogenetic analyses of the SCAU-IFDC01 fungus isolate from *D. citri* based on ITS1/ITS5 and maximum likelihood method.

**Figure 3 insects-13-00374-f003:**
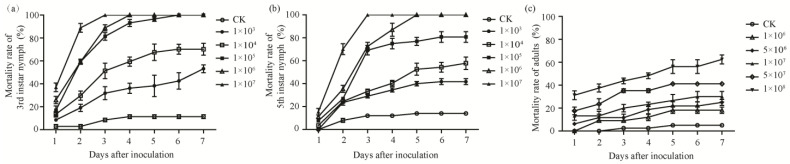
Pathogenicity of *C. fumosorosea* SCAU-CFDC01 against *D. citri*. (**a**–**c**) Trend over time in the percentage mortality of 3rd and 5th instar nymphs and adults of *D. citri* with days after inoculation with six concentrations of *C. fumosorosea*. CK (control check): distilled water.

**Figure 4 insects-13-00374-f004:**
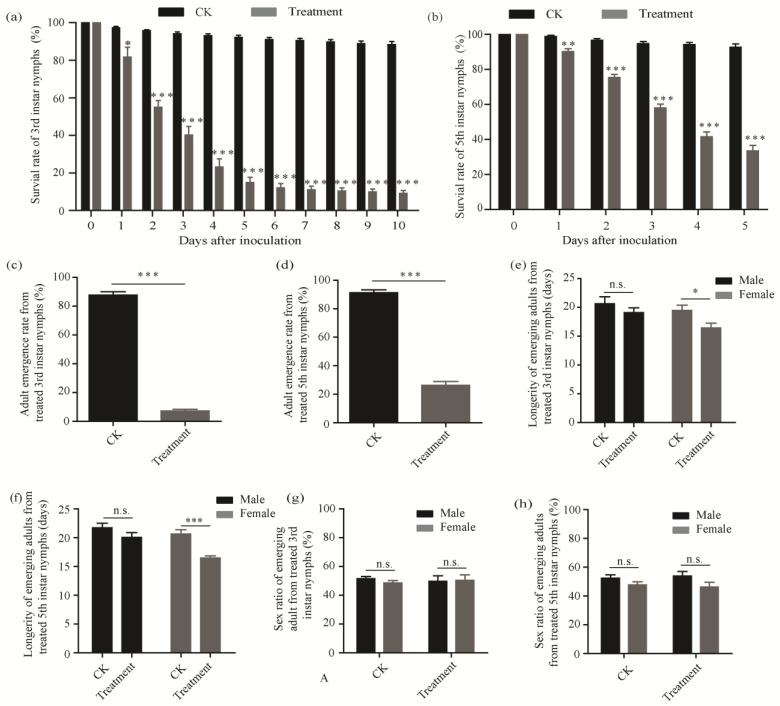
The effects of *C. fumosorosea* SCAU-CFDC01 on *D. citri* adults from nymphs treated. Adult survival rate (**a**), emergence rate (**c**), longevity (**e**), and sex ratio (**g**) based on a 3rd instar nymph cohort inoculated with the fungus. Adult survival rate (**b**), emergence rate (**d**), longevity (**f**), and sex ratio (**h**) based on a 5th instar nymph cohort inoculated with the fungus. All the results are presented as the mean ± standard error (SE). *t*-test statistically significant differences between treatment and CK were denoted by * (*p* < 0.05), ** (*p* < 0.01), and *** (*p* < 0.001). n.s.—not significant.

**Table 1 insects-13-00374-t001:** Fungal species, GenBank accession number, isolate number, location, number of nucleotides analyzed, and of entomopathogenic fungi.

Species	GenBank Accession Number	Isolate Number	Location	Number of Nucleotides Analyzed (bp)
*C*. *fumosorosea*	OL872288.1	SCAU-CFDC01	Guangzhou, China	564
*C*. *fumosorosea*	MG837716	FAFU-1	Fuzhou, China	603
*C*. *fumosorosea*	HM209050.1	ARSEF:3302	Mexico	570
*C*. *fumosorosea*	HM209049.1	CNRCB1	Mexico	572
*C*. *fumosorosea*	FJ765017.1	SKCH-1	Guangzhou, China	635
*C*. *fumosorosea*	KX057373.1	IfTS02	Guangzhou, China	564
*M*. *anisopliae*	KX380792.1	YD2-1-8	Nanchang, China	559
*M*. *anisopliae*	JN377427.1	zj	Wenchang, China	685
*B*. *brongniartii*	JX110381.1	SASR HHB32B	South Africa	595
*B*. *brongniartii*	JF947191.1	ARSEF8153	Canada	593
*B*. *bassiana*	MH854860.1	CBS 119.26	Indonesia	637
*B*. *bassiana*	KF772861.1	YD-1	Wuhan, Hubei	491
*B*. *bassiana*	AY531961.1	813	USA	797
*Beauveria pseudobassiana*	LT220765.1	TM1527B	Portugal	958
*I*. *amoenerosea*	HM161732.1	CBS729.73	Thailand	865

**Table 2 insects-13-00374-t002:** Pathogenicity regression equations for LC_50_ values of *C. fumosorosea* SCAU-CFDC01 against *D. citri*.

Developmental Stage	Days	Slop ± SE	Regression Virulence Model	LC_50_ (Conidia mL^−1^) (95% CI)	*p* *
3rd instar nymphs	3	0.66 ± 0.09	*Y* = 0.66*X* − 2.51	6.0 × 10^3^ (2.37 × 10^3^–1.25 × 10^4^)	0.65
	4	0.72 ± 0.11	*Y* = 0.72*X* − 2.49	3.00 × 10^3^ (7.73–1.86 × 10^4^)	0.06
	5	1.03 ± 0.16	*Y* = 1.03*X* − 3.46	2.32 × 10^3^ (1.10 × 10^3^–4.08 × 10^3^)	0.72
	6	1.16 ± 0.19	*Y* = 1.16*X* − 3.77	1.82 × 10^3^ (8.94 × 10^2^–3.09 × 10^3^)	0.25
	7	0.99 ± 0.18	*Y* = 0.99*X* − 3.03	1.14 × 10^3^ (1.61–4.63 × 10^3^)	0.11
5th instar nymphs	3	0.56 ± 0.28	*Y* = 0.56*X* − 2.84	1.14 × 10^5^ (2.15 × 10^4^–8.04 × 10^5^)	0.17
	4	0.86 ± 0.26	*Y* = 0.86*X* − 4.13	6.55 × 10^4^ (1.97 × 10^3^–2.15 × 10^5^)	0.17
	5	1.63 ± 0.42	*Y* = 1.63*X* − 7.28	2.86 × 10^4^ (6.31 × 10^3^–5.84 × 10^4^)	0.51
	6	1.74 ± 0.53	*Y* = 1.74*X* − 7.92	3.46 × 10^4^ (4.76 × 10^3^–7.23 × 10^4^)	0.54
	7	1.57 ± 0.44	*Y* = 1.57*X* − 6.98	2.88 × 10^4^ (4.08 × 10^3^–6.42 × 10^4^)	0.44
Adult	3	0.32 ± 0.08	*Y* = 0.32*X* − 2.36	2.68 × 10^7^ (3.25 × 10^6^–6.73 × 10^9^)	0.96
	4	0.30 ± 0.08	*Y* = 0.30*X* − 2.18	1.85 × 10^7^ (2.26 × 10^6^–5.01 × 10^9^)	0.98
	5	0.27 ± 0.08	*Y* = 0.27*X* − 1.84	5.43 × 10^6^ (7.73 × 10^5^–7.51 × 10^8^)	0.87
	6	0.27 ± 0.08	*Y* = 0.27*X* − 1.82	4.76 × 10^6^ (6.89 × 10^5^–6.02 × 10^8^)	0.94
	7	0.26 ± 0.08	*Y* = 0.26*X* − 1.70	4.69 × 10^6^ (6.38 × 10^5^–8.82 × 10^8^)	0.95

*p* * goodness of fit test. All *p* > 0.05 showed good regression fit to the probit model.

**Table 3 insects-13-00374-t003:** Pathogenicity regression equations for LT_50_ values of *C. fumosorosea* against *D. citri*.

Developmental Stage	Conidia Concentration	Slop ± SE	Regression Model	LT_50_ (days) (95% CI)	*p*
3rd instar nymphs	1 × 10^7^	5.71 ± 1.10	*Y* = 5.71*X* − 0.36	1.18 (0.94–1.35)	0.09
	1 × 10^6^	4.55 ± 0.65	*Y* = 4.55*X* − 0.85	1.53 (1.26–1.78)	0.95
	1 × 10^5^	4.20 ± 0.38	*Y* = 4.23*X* − 1.11	1.75 (1.54–1.94)	0.91
	1 × 10^4^	2.01 ± 0.32	*Y* = 2.06*X* − 1.06	3.28 (2.69–3.94)	0.64
	1 × 10^3^	1.58 ± 0.30	*Y* = 1.58*X* − 1.33	6.93 (5.40–10.85)	0.10
5th instar nymphs	1 × 10^7^	6.35 ± 0.58	*Y* = 6.35*X* − 1.14	1.51 (1.39–1.63)	0.31
	1 × 10^6^	4.78 ± 0.39	*Y* = 4.78*X* − 1.58	2.14 (1.75–2.51)	0.06
	1 × 10^5^	3.32 ± 0.32	*Y* = 3.32*X* − 1.42	2.67 (2.34–2.97)	0.40
	1 × 10^4^	2.10 ± 0.29	*Y* = 2.10*X* − 1.48	5.09 (4.39–6.17)	0.77
	1 × 10^3^	1.75 ± 0.31	*Y* = 1.75*X* − 1.54	7.60 (6.03–11.43)	0.17
Adults	1 × 10^8^	1.61 ± 0.58	*Y* = 1.61*X* − 0.13	1.20 (0.00–3.67)	0.74
	5 × 10^7^	1.57 ± 0.64	*Y* = 1.57*X* − 0.56	2.28 (0.00–5.89)	0.97
	1 × 10^7^	1.63 ± 0.62	*Y* = 1.63*X* − 0.96	3.88 (0.14–6.99)	0.88
	5 × 10^6^	1.24 ± 0.54	*Y* = 1.24*X* − 1.01	6.51 (0.04–13.78)	0.83
	1 × 10^6^	1.28 ± 0.83	*Y* = 1.28*X* − 1.26	9.71 (0.21–15.78)	0.98

## Data Availability

The data presented in this study are available in article.

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
