# Peer review of "Identification of a Cordyceps fumosorosea Fungus Isolate and Its Pathogenicity against Asian Citrus Psyllid, Diaphorina citri (Hemiptera: Liviidae)"

_insects, 2022, doi:10.3390/insects13040374_

Round 1
Reviewer 1 Report
The main strength of this research is the series of greenhouse and laboratory experiments. The results validate previous studies and provide interesting insights. However, the statistical analyses require some important considerations and interpretations. The genomic approach was useful for fungus identification. The phylogenetic analysis provides, however, little information. The morphological section also requires additional work.
The document in the status actual cannot be published in the MDPI Journal. It requires a major and careful review of concepts, methods, interpretation, and English language usage. All comments are directly included in the manuscript for the authors' consideration.
Finally, a philosophical consideration. The potential of entomopathogens for D. citri suppression and many other arthropods is unquestionable. The functional response is a trophic system naturally established. However, to assure the application of such organisms at the agricultural level requires comprehensive studies associated with the ‘search’ for potential beneficial parasites: regional exploring, seasonal effects, agricultural systems, the host range of cultivars, climatic considerations, insect population dynamics, phenology characterization, etc. However, we continue developing classical work focusing on scouting individual parasitized insects. To succeed, we must enhance our work to another level. Otherwise, we will continue reporting only possibilities feasible under controlled conditions. In addition, we need to move to a new rational frame for biological control. LD50, LT50, toxicity, mycopesticides, biopesticides, etc. are terminologies borrowed from the classical insecticide approach with philosophical, conceptual, and operative drawbacks in biological control.

Author Response
L1: “A” should be “a ”, delete “new”
L2: “Toxicity Evaluation” should be “pathogenicty”
L4: “Psyllidae” should be “Liviidae”
L12: “of citrus Huanglongbing” should be “Clas”, “a” should be “an”, delete “new”
L13: Delete “strain”, “The isolate” should be “ The isolate was identified as”
L14: “through morphological identification and ITS sequence” should be “base on morphology and ITS”
L17: “ virulent” should be “pathogenic”
L18: “of” should be “on”, “virulence” should be “pathogenicity”
L21-22: “D. citri, and these findings will facilitate mycopesticide development for biological control of D. citri.” Should be “D.citri. These findings may provide insights for potential biological control programs toward D. citri under field conditions.”
L37: Delete “Diaphorina citri; Cordyceps fumosorosea; pathogenicity; ”
L45: “twig dieback ” is not specific to this disease. Please remove it.
L56: Change “to kill” to “for”
L63: “with” not required
L71: “ B. bassiana, M. anisopliae” should be “ B. bassiana, M. anisopliae”
L73: “strains” should be “isolates”, “the” shoule be “a”
L75: “strains” should be “isolates”
L78: “isolated 12 fungal strains” should be “tested 12 fungal isolates”
L79: “found C. javanica” should be “found that C. javanica”
L80: “strain” should be “isolate of”, “isolated” should be “cultured”
L83: “to D. citri larvae and adults” should be “to nymphs and adults of D. citri”
L91: “areas” should be “area”
L93-94: “on the water” should be “on water”
L112: Require space
L122: “fungal” should be “fungi”
L123: “An” should be “a”
L133: “Genbank number” should be “GenBank accession number”
L138: “M. paniculata” should be “M. paniculata”
L139: “colony” should be “plant-colony”
L174: “M. paniculata” should be “M. paniculata”
L177: “sex” should be “and sex”
L182: “mortality” should be “insect mortality rate”, “one-way ANOVA” should be “Split plot ANOVA”
L184: “derived” should be “derived from”
L186: “Survival” should be “survival”
L193: “day” should be “15 days of”
L196: Delete “to”
L201: “nymphs” should be “nymph”
L203: “nymphs” should be “nymph”
L207: “homologies” should be “homology”, “strains” should be “isolates”
L207-L209: Delete “C. fumosorosea”
L210: “strain C. fumosorosea SCAU-CFDC01.” Should be “C. fumosorosea isolate SCAU-CFDC01 with the genbank accession number OL872288.1 .”
L214: “Virulence” should be “Pathogenicity”
L216: “days” should be “time”
L218-L223: Delete “F = 101.9; df = 5, 72; F = 260.5; df = 5, 72; F = 55.5; df = 5, 72; F = 520.0; df = 6, 72; F = 596.0; df = 6, 72; F = 55.33; df =6, 72; F = 26.62; df = 30,72; F = 32.42; df = 30, 72; F = 2.17; df = 30, 72”
L214: “Virulence” should be “Pathogenicity”
L216: “days” should be “time”
L218-L223: Delete “F = 101.9; df = 5, 72; F = 260.5; df = 5, 72; F = 55.5; df = 5, 72; F = 520.0; df = 6, 72; F = 596.0; df = 6, 72; F = 55.33; df =6, 72; F = 26.62; df = 30, 72; F = 32.42; df = 30, 72, F = 2.17; df = 30, 72”
L237: “1.16” should be “1.18”, “9.71” should be “9.72”, “107” should be “108”
L243: Delete “virulence”
L245-L246: “adults emerging from treated 3rd and 5th instar nymphs” should be “adults from 3rd and 5th instar nymphs treated”
L247-248: “adults emerging from treated 3rd instar nymphs: t = 101.8; df = 5; p < 0.001, adults emerging from treated 5th instar nymphs: t = 17.21; df = 5; p < 0.001” should be “p < 0.001”
L249: “strain” should be “isolate”
L250: “were” should be “are”
L251: Delete “emerging”
L253-L254: Delete “adults emerging from treated 3rd instar nymphs: t = 26.60; df = 5; p < 0.001, adults emerging from treated 5th instar nymphs: t = 26.85; df = 5”
L256: Delete “t = 2.33; df = 41”
L257: Delete “t = 5.45; df = 89”
L272: “statistically” should be “T-test statistically”
L275: “the new isolate” should be “a new fungus isolate from D. citri”
L278: “Cordyceps spp” should be “C. fumosorosea”
L279: “ITS” should be “ITS1/ITS5”
L280: “showed” should be “shown”
L281: Delete “a new”, “toxicity” should be “pathogenicity”
L282: “with respective” should be “resulting”
L283: “separately” should be “respectively”, “B. bassiana” should be “B. bassiana”
L285-L286: “B.bassiana and M. anisopliae” should be “B. bassiana and M. anisopliae”
L287: Delete “fungus”, “of”
L288: Delete “C. javanica”
L289: “separately” should be “respectively”
L292: “high quality strain” should be “other fungus”
L300: Delete “method lead to difference in fungal virulence”
L301: “fungi ” should be “fungi is”
L302: “affect” should be “affected”, “temperature etc” should be temperature, agricultural practices, etc”
L304: “vitally important” should be “stages and environmental conditions is important. Validate at field conditions is also fundamental.”
L311: “virulence” should be “pathogenic”
L313: “HLB” should be “Clas”
L319: “good control” should be “potential good control”
L320: “reluts” should be “findings”
L321: “fantastic” should be “excellent prospects”
L323: “entomogenous” should be “entomopathogenic”
L326: “its highly virulent effect of D. citri.” should be “its effect on the mortality, survival, and emergence rate, as well as on the longevity, and sex ratio of D. citri”
L327: Delete “preparation in”
L328: Delete “comprehensive”
L357: “Diaphorina citri” should be “Diaphorina citri”
Response: Thanks very much for your great suggestion. We have revised the main text as the editor’s suggestions. Revised portions are marked with “track changes” in our new revised manuscript.
L42: Please modify the emphasis on the losses' impact.
Response: Thanks very much for your great suggestion. We have rewritten these sentences, deleted the Bove’s papers, and added the new reference (DOI 10.1007/s10658-011-9779-1).
L47: Please leave italics only on the genus. Also, do that for the abbreviated form. Maintain that on the whole document.
Response: Thanks very much for your great suggestion.
L55: Review the English language
Response: Thanks very much for your great suggestion. We have reviewed.
L59:“There is another important reason.Consider this: '....also insecticides are restricted for urban areas where D. citri may reproduce on ornamental (i.e. Murraya paniculata) and backyard citrus plants”
Response: Thanks very much for your great suggestion. We have added it.
L69-70: Abbreviate genus since it was previously referred. i.e. C.
Response: Thanks very much for your great suggestion.
L76: “review the journal style”
Response: Thanks very much for your great suggestion. We have reviewed and checked.
L86: mycoparasite is more appropriate for the scope of this research
Response: Thanks very much for your great suggestion. We have revised it.
L90: recorder in (include date or season)
Response: Thanks very much for your great suggestion. We have added the content.
L97: A pure culture can only be assured with nonosporic culture. This was not the case. Therefore all we can say here is:....' to get the fungal isolate'
Response: Thanks very much for your great suggestion. We have revised it.
L99: “and inoculation, The heading section is better described, Authors are inferring from the morphological characterization the fungus identification. This is not the case in the way the data is presented in this document. To justify the latter, the authors should include the taxonomic references used for identification purposes and indicate which the main morphological features were.”
Response: Thanks very much for your great suggestion. We have added the content.
L104: You may removed “the”
Response: Thanks very much for your great suggestion. We have removed it.
L105: Review according to journal indications.
Response: Thanks very much for your great suggestion. We have revised them.
L106: You may consider eliminate. Not required
Response: Thanks very much for your great suggestion. We have deleted it.
L107: Indentification and phylogenetic ......of fungal isolate. Alternative heading
Response: Thanks very much for your great suggestion. We have revised it.
P108: This may be more appropiate: 'The mycelia was harvested´
Response: Thanks very much for your great suggestion. We have revised it.
L111: At this point, we do not know which are the 'all fungi'...The authors are not clear on this.
Response: Thanks very much for your great suggestion. We have rewritten it.
L121: “Alignment” should be “aligned”, The author must register their fungus isolate sequence in the GenBank. If so, provide the accession number.
Response: Thanks very much for your great suggestion. We have provided the accession number.
L134: “Assure some order.e.g. all Beauveria must be together. Strain number should be Isolate number.”
Response: Thanks very much for your great suggestion. We have revised them.
L136: how many?
Response: Thanks very much for your question. The number of plant is 300.
L143: “larvae” should be “nymphs”, eliminate “nymphs”; Please clarify: Adults cannot be ready in 1-2 days after eclosion.
Response: Thanks very much for your question. We have revised them.
“adults (1-2 days days after eclosion)” should be “newly-emerged adults (1-2 days old)”.
L144: how many plantas?
Response: Thanks very much for your question. The number is nine.
L165: and complete Koch´s postulates. Authors actually completed the postulates?
Response: Thanks very much for your question. We have completed the Koch´s postulates.
L169: seedling or flushes? No clear.
Response: Thanks very much for your question. We refer to flushes and have revised them.
L176: “until” should be “from” ; It is important to clarify. Counting from emergence or inoculation? If both, results should be clear on that. e.g. biological survival is different from infection survival.
Response: Thanks very much for your question. We counted both from emergence and inoculation and have revised them.
L192: the ring pattern is not shown
Response: Thanks very much for your question. We have rewritten it.
L194: picture contrast is too bright to see the 'rose' color. Compare with Fig. 1f, This picture can be enhanced by enlarching and trimming the picture border. The idea is to have a better view of spores´ clavete shape. It is not clear.
Response: Thanks very much for your question. We have revised them.
L198: Be consistent....in Figure 1´s caption authors are referring to: 3rd day, 5th, and 9th days after infection. Also, authors should be aware that inoculation, infection, and colonization are different processes. Here the most appropriate is 3rd day, 5th, and 9th days after inoculation. The authors are not sure when the actual infection took place. Similar correction applies for other references on this. “hyphae” should be “mycelia”
Response: Thanks very much for your question. We have rewritten them.
L205: The authors described some fungus morphological features but they were not related to any taxonomic reference. Also, inoculation per se does not provide any evidence of the species' identity.
Response: Thanks very much for your suggestion. We have added the taxonomic reference and rewritten them.
L206: Genomic analyses further... Note: See the previous comment on this.
Response: Thanks very much for your suggestion. We have revised it.
L212: the SCAU-IFDC01 fungus isolate from D. citri based on ITS1/ITS5 and maximum likelihood method. Consider this Figure caption instead: Molecular identification and phylogenetic analyses of the SCAU-IFDC01 fungus isolate from D. citri based on ITS1/ITS5 and maximum likelihood method.
Response: Thanks very much for your suggestion. We have rewritten it.
L223: The assay was repited three times...how the authors manage the statistical results of this series of experiments. In split plot desing when the interaction is significative then individual factors effects are ruleout. Authors should clarify the meaning of the time effect on the mortality upom the different concentrations. Otherwise there is not much use of this analysis.
Response: Thanks very much for your suggestion. We have revised them.
L224: Standardize the bullets style for the same concentration in all population cohorts
Response: Thanks very much for your suggestion. We have revised it.
L230: “old nymphs” should be “5th instar”, Authors performed two replications of this experiment. The idea was to measure the consistence on the results. Then it should be discused how the LC50 and LT50 performed on these two experiments.
Response: Thanks very much for your suggestion. We have revised it.
L232: Delete “virulence”; clarify its meaning
Response: Thanks very much for your suggestion. We have deleted it and clarified the P meaning.
L258: This statement needs further clarification. Results on Figure 4 are perfectly clear.
Response: Thanks very much for your suggestion. We have rewritten it.
L267: Adult survival rate (a), emergence rate (c) longevity (e), and sex ratio (g) based on a 3rd instar nymph cohort inoculated with the fungus. Note: I suggest using this structure for the counterpart 5th instar. For b,d,f and h descritption see previous note.
Response: Thanks very much for your suggestion. We have rewritten them.
L273: add “n.s. is not significant.”
Response: Thanks very much for your suggestion. We have added it.
L276: Author should provide taxonomic evidence on the Results section.
Response: Thanks very much for your suggestion. We have provided the taxonomic evidence in the Results section .
L293: Include at the end the following statement: 'under controlled conditions'. Note. It is important to emphasize that these results and others are under controlled conditions. Validation under field condition requires to be made.
Response: Thanks very much for your suggestion. We have rewritten it.
L308: Consider to include as another statement: 'In addition, lethality requires an infection period, therefore an infected adult may be able to transmit CLas before its eventual death'.
Response: Thanks very much for your suggestion. We have added it.
L320: Include these references: 16, 17, 20
Response: Thanks very much for your suggestion. We have added these.
L344: Include this number on the main body text of the document.
Response: Thanks very much for your suggestion. We have revised it.
L349: Verify standardization on Journal Name citation.
Response: Thanks very much for your suggestion. We have verified standardization and rewritten it.
L386: Verify standardization on Journal Name citation.
Response: Thanks very much for your suggestion. We have revised it.
L389: Verify standardization on Journal Name citation.
Response: Thanks very much for your suggestion. We have checked it.
Reviewer 2 Report
This ms is within the scope of journal, the authors isolated a virulent isolate of Cordyceps fumosorosea from the cadaver of Asian citrus psyllid (Diaphorina citri). The fungal isolate was identified both on morphological and ITS sequence basis. The pathogenicity of C. fumosorosea isolate was evaluated in laboratory and greenhouse against nymphs and adults of D. citri. The results showed that the isolate was effectve against both nymps and adults. In greenhouse the high level of virulence against nymphs was noted, the adults emergence and female longevity was also effected.
This is a comprehensive and well-written study, the objectives of the study are clear, the experimental design is appropriate and the results support the conclusion. This is a well-written ms and authors generated novel data which convinced me to accept this research ms for its publication in Insects with some minor changes in the ms
Some other concerns are as follow:
L71; italicize scientific name/s of fungi here and all the scientific names throughout ms
L73; write complete genus name of each scientific name at start of sentence/paragraph throughout ms
L99; how you confirmed the morphological characteristsics of fungi, cite the relevant taxonomic keys?
L181; pl clarify, did you normalize the data before analysis? give reference for “Tukey-Kramer HSD test” and how the mortality was corrected, give reference?
L274; it is better the Discussion section should be more strengthened – the authors should provide the findings of few more relevant studies and then compare them logically with possible reasoning of variation in the other’s findings with the results of the current study
Author Response
L71: italicize scientific name/s of fungi here and all the scientific names throughout ms
Response: Thanks very much for your great suggestion. We have italicized scientific name of fungi and scientific name throughout ms.
L73: write complete genus name of each scientific name at start of sentence/paragraph throughout ms
Response: Thank you very much for your suggestion. We have revised them.
L99: how you confirmed the morphological characteristsics of fungi, cite the relevant taxonomic keys?
Response: Thank you very much for your suggestion. We have cited the relevant taxonomic keys.
L181: pl clarify, did you normalize the data before analysis? give reference for “Tukey-Kramer HSD test” and how the mortality was corrected, give reference?
Response: Thank you very much for your suggestion. We had normalized the data before analysis, and added the reference for “Tukey-Kramer HSD test”.
L274: it is better the Discussion section should be more strengthened – the authors should provide the findings of few more relevant studies and then compare them logically with possible reasoning of variation in the other’s findings with the results of the current study
Response: Thank you very much for your suggestion. We have revised them.
Reviewer 3 Report
The authors investigate the fungal entomopathogenic activity against Asian Citrus Psyllid, (Hemiptera: Psyllidae). They characterize the fungal isolate and studied its entomopathogenic activity to D. citri larvae (3rd and 5th instar nymphs), and adults (1-2 days after eclosion). it's an original article with nice and clear presentation results.
The manuscript was written in understandable English.
Therefore, I recommend this manuscript be published in the Journal Insects.
Minor comments:
1) Table 1: Could you please add "Sequence identity (%)" and "Number of nucleotides analyzed (bp)" to Table 1.
2) A sentence describing Koch’s postulates must be added to the Material and Methods section. How would you prove that a particular organism was the cause of the entomopathogenic activity? To be sure that you have the right fungus and don't confuse it with another one, a small piece of mycelium should be transferred to a fresh fungal medium. The obtained fungus was determined by microscopy, which conforms to your isolate.
Author Response
1) Table 1: Could you please add "Sequence identity (%)" and "Number of nucleotides analyzed (bp)" to Table 1.
Response: Thank you very much for your great suggestion. We have added them.
2) A sentence describing Koch’s postulates must be added to the Material and Methods section. How would you prove that a particular organism was the cause of the entomopathogenic activity? To be sure that you have the right fungus and don't confuse it with another one, a small piece of mycelium should be transferred to a fresh fungal medium. The obtained fungus was determined by microscopy, which conforms to your isolate.
Response: Thank you very much for your great suggestion. We have added Koch’s postulates in the Material and Methods section.
Reviewer 4 Report
I reviewed the manuscript entitled “Identification of A New Cordyceps fumosorosea Fungus Isolate and Its Toxicity Evaluation against Asian Citrus Psyllid, Diaphorina citri (Hemiptera: Psyllidae)”, aimed to evaluate the pathogenicity of Cordyceps fumosorosea that authors isolated against D. citri nymphs and adults. Although the topic is interesting and the results seem encouraging, the manuscript presents many issues that the authors should clarify and resolve. For example, there are several sentences that are missing words or are written in an unclear manner. Therefore, I strongly suggest that the authors make a thorough revision of the ms in collaboration with a native speaker in order to overcome the impression of superficiality that is particularly noticeable in some sections of the ms. This impression is amplified by the fact that in the text the scientific name of the various animal or plant species is not in italics
Some considerations
The introduction is confusing in some parts and should therefore follow a clearer logical thread. Between lines 46 and 58, for instance, the topics D. citri and HLB disease alternate in a non-logical and not very exhaustive way.
Figure 3 is, in my opinion, completely unreadable, perhaps because of the size or the choice of symbols that complete the curves.
In Figure 1 authors should show the upper side of the colony on SDAY on 5th day and the back side of the colony on SDAY on 13th day
The discussion of the results should be improved and enhanced beginning with the first sentence which is truncated.
Other observations
R 43 Industry?
R 49 It is the first time that the species has been mentioned so it is necessary to write Diaphorina citri
R 50 Discovered or described instead of reported
R 58 delete “et al”
R 61/63 Reformulate
R 67/68 Reformulate
R 71 rewrite “after”
R 81 delete the dot
R 84 delete the dot
R 90 I do not understand if it was an occasional collection, or the authors wanted to explore the possibility to find entomopathogens in a D. citri population considered.
R 94 Were the cadavers incubated at 4° C?
R 95 After the 24 hours, what was the temperature of incubation?
R 168/169 M. paniculata plants with about ten fresh seedlings. What does it mean?
R 191 The title of this paragraph is “Identification of the isolate based on morphological characteristics and phylogenetic analysis” but the authors refer the results of inoculation on D. citri specimens.
R 193 “after 15” days, hours? Actually in the figure (1 b) authors report the growth of fungus after 13 days and not 15.
R 196/197 Infection for 48h. I do not understand this
R 228 Conidia
R 205 It is not the inoculation that determines the fungus!
R 321 I understand the enthusiasm, but "fantastic" seems a bit of an overstatement.
Author Response
The introduction is confusing in some parts and should therefore follow a clearer logical thread. Between lines 46 and 58, for instance, the topics D. citri and HLB disease alternate in a non-logical and not very exhaustive way.
Response: Thanks very much for your great suggestion. We have rewritten these sentences.
Figure 3 is, in my opinion, completely unreadable, perhaps because of the size or the choice of symbols that complete the curves.
Response: Thanks very much for your great suggestion. We have standardized the bullets style for the same concentration in all population cohorts.
In Figure 1 authors should show the upper side of the colony on SDAY on 5th day and the back side of the colony on SDAY on 13th day
Response: Thanks very much for your great suggestion. Firstly, figure 1a showed the upper side of the colony on SDAY on 5th day. Secondly, SDAY medium is the best medium for SCAU-CFDC01 isolate. Since medium is brown, the back sides of the colony on SDAY is very unclear. So we only show the upper sides of the colony on SDAY.
The discussion of the results should be improved and enhanced beginning with the first sentence which is truncated.
Response: Thanks for your suggestion. We have rewritten these sentences.
Other observations
R43 Industry?
Response: Thanks very much for your great suggestion. We have revised it.
R49 It is the first time that the species has been mentioned so it is necessary to write Diaphorina citri
Response: Thanks very much for your great suggestion. We have revised it.
R50 Discovered or described instead of reported
Response: Thanks very much for your great suggestion. We have revised it.
R58 Delete “et al”
Response: Thanks very much for your great suggestion. We have deleted it.
R61/63 Reformulate
Response: Thanks very much for your great suggestion. We have reformulated it.
R67/68 Reformulate
Response: Thanks very much for your great suggestion. We have reformulated it.
R71 rewrite “after”
Response: Thanks very much for your great suggestion. We have reformulated it.
R81 delete the dot; R 84 delete the dot
Response: Thanks very much for your great suggestion. We have deleted it.
R90 I do not understand if it was an occasional collection, or the authors wanted to explore the possibility to find entomopathogens in a D. citri population considered.
Response: We wanted to explore the possibility to find entomopathogens in a D. citri population considered. We collected a lot of D. citri in the field and expect to find the highly pathogenic fungi for further biological control.
R94 Were the cadavers incubated at 4° C?
Response: Yes, the cadavers were incubated at 4 ℃.
R 95 After the 24 hours, what was the temperature of incubation?
Response: The cadaver was incubated on water medium in an incubator (4 ± 1℃, 65 ± 5% RH (relative humidity) with total darkness) until mycelia or visible spores observed.
R168/169 M. paniculata plants with about ten fresh seedlings. What does it mean?
Response: D. citri only lay eggs when plants have flushes. In order to ensure the consistency of the experimental conditions, we trimmed the plants, leaving the same flushes on each plant.
R191 The title of this paragraph is “Identification of the isolate based on morphological characteristics and phylogenetic analysis” but the authors refer the results of inoculation on D. citri specimens.
Response: Thanks very much for your great suggestion. We have revised the title.
R193 “after 15” days, hours? Actually in the figure (1 b) authors report the growth of fungus after 13 days and not 15.
Response: Thank you very much for your question. There should be “after 15 days”, and we have corrected the error in the figure (1 b).
R196/197 Infection for 48h. I do not understand this
Response: Thank you very much for your question. “Infection for 48h” should be “After 48h of inocubation”
R228 Conidia
Response: Thanks very much for your great suggestion. We have revised it.
R205 It is not the inoculation that determines the fungus!
Response: Thanks very much for your great suggestion. We have revised it.
R321 I understand the enthusiasm, but "fantastic" seems a bit of an overstatement.
Response: Thanks very much for your great suggestion. We have changed “fantastic” to “excellent prospects”
Round 2
Reviewer 1 Report
I believe the document is now suitable for publication after some minor corrections. Thank you for improving it.
Here are some important considerations; others are directly indicated in the document:
- Be sure of the right writing for CLas, CLam, CLaf
- Objective 1 needs rewriting since 'diversity' cannot be supported based on the limited field sampling. I forgot to mention this in my first review. Sorry.
- Due to the significance of the interaction on the Split Plot ANOVA design, this result is the one to be addressed in the document. It is in agreement with the overall results.
- Please standardize the references. In particular, some Journal names end with a period or comma. Some do not apply.

Author Response
1. Be sure of the right writing for CLas, CLam, CLaf
Response: Thanks very much for your suggestion. We have rewritten them.
2. Objective 1 needs rewriting since 'diversity' cannot be supported based on the limited field sampling. I forgot to mention this in my first review. Sorry.
Response: Thanks very much for your suggestion. We have revised it.
3. Due to the significance of the interaction on the Split Plot ANOVA design, this result is the one to be addressed in the document. It is in agreement with the overall results.
Response: Thanks very much for your suggestion. We have revised it.
4. Please standardize the references. In particular, some Journal names end with a period or comma. Some do not apply.
Response: Thanks very much for your suggestion. We have verified standardization and rewritten it.
L38: Consider to adding: HLB, Citrus Greening, Entomopathogen
Response: Thanks very much for your suggestion. We have added them.
L34: “significantly” should be “was significantly"
L42: “destructive” should be “ severe”
L55: “orn amental” should be “ornamental”, “back yard” should be “backyard”
L56: “resistance to” should be “resistance to at least”
L66: “discovered” should be “reported”
L79: “the molecular and morphological diversity of the isolate” should be “the isolate using molecular and morphological characteristics”
L178: “three” should be “three times”
L179: “2021 autumn” should be “autumn 2021”
L219-L222: “by conidia concentration(3rd instar nymphs: p < 0.001, 5th instar nymphs: p < 0.001, adults: p < 0.001) and days after inoculation(3rd instar nymphs: p < 0.001, 5th instar nymphs: F=p < 0.001, adults: p < 0.001) should be “by the interaction of conidia concentration and days after inoculation (3rd instar nymphs, and 5th instar, p < 0.001; adults, p < 0.05), meaning that inoculum load effect was dependent of the exposure time.
L245: Delete “treated”
L276: “strong” should be “severe”
L277: Delete “of C. javanica”
L296: “Nymphs” should be “Nymphs”
L301: “Clas” should be “CLas”
L306: “Clas” should be “CLas”
L325: “to further identify by transcriptome analysis” should be “further identification by transcriptome analysis or other omic strategy”
L334: Delete extra space after “alternative”
L363: “Diaphorina citri” should be “Diaphorina citri”
L389: “AlatorreRosas” should be “Alatorre-Rosas”
L395: “Diaphorina citri” should be “Diaphorina citri”
L424: “MoraAguilera” should be “Mora-Aguilera”
L438: “RodríguezGuerra” should be “Rodríguez-Guerra”
Response: Thanks very much for your suggestion. We have revised the main text as the reviewer’s suggestions. Revised portions are marked with “track changes” in our new revised manuscript.
Reviewer 4 Report
I am satisfied with the answers of the authors to my comments.
Author Response
Thank you very much for your approval.